

# An adaptive cryptosystem on a Finite Field

Awnon Bhowmik[1] and Unnikrishnan Menon[2]

[1] Department of Mathematics, The City College of New York, New York, NY, United States of America
[2] Department of Electrical and Electronics Engineering, Vellore Institute of Technology University, Vellore, Tamil Nadu, India

## ABSTRACT

Owing to mathematical theory and computational power evolution, modern cryptosystems demand ingenious trapdoor functions as their foundation to extend the gap between an enthusiastic interceptor and sensitive information. This paper introduces an adaptive block encryption scheme. This system is based on product, exponent, and modulo operation on a finite field. At the heart of this algorithm lies an innovative and robust trapdoor function that operates in the Galois Field and is responsible for the superior speed and security offered by it. Prime number theorem plays a fundamental role in this system, to keep unwelcome adversaries at bay. This is a self-adjusting cryptosystem that autonomously optimizes the system parameters thereby reducing effort on the user's side while enhancing the level of security. This paper provides an extensive analysis of a few notable attributes of this cryptosystem such as its exponential rise in security with an increase in the length of plaintext while simultaneously ensuring that the operations are carried out in feasible runtime. Additionally, an experimental analysis is also performed to study the trends and relations between the cryptosystem parameters, including a few edge cases.

## INTRODUCTION

Cryptography is the art of hiding messages to provide it with a certain level of security to maintain confidentiality and integrity. This new idea, whether it was to hide secret messages, or to transform the original message to make it look fancy, dignified, etc. continued through the medieval ages, the renaissance period saw the birth of the polyalphabetic substitution cipher, called the Vigenère Cipher (*Rubinstein-Salzedo, 2018*). An encryption device called the Enigma machine (*Singh, 1999*) was used by the Nazi Germans during World War II. Although history suggests that it has been in use for ages, systematic study of cryptology as a science (and perhaps an art) just started around one hundred years ago (*Sidhpurwala, 2013*).

But it was not until the 1970s, that studies in cryptography got serious. Data Encryption Standard (DES) was introduced by IBM in 1976 (*Tuchman, 1997*) followed by Diffie Hellman Key Exchange in the same year (*Kallam, 2015*). In 1977, RSA came along (*Calderbank, 2007*) and in 2002, AES was accepted as a standard security protocol

Corresponding author
Awnon Bhowmik,
abhowmik901@york.cuny.edu

to be used in both hardware and software (*Dworkin et al., 2001*). And thus cryptography became popular.

The strength or foundation of a modern encryption protocol relies upon the inherent Trapdoor Function. As classical cryptography evolved, it has become clear that some key components are essential in making stronger trapdoor functions, also known as one-way functions. Studies have shown that prime numbers are an essential part of numerous cryptosystems, and with a bit of effort, numerous mathematical concepts can be used to generate stronger cryptosystems. Conventional, widely used algorithms such as RSA rely on integer products involving large primes. Breaking this system is essentially an attempt to solve the integer factorization problem, which can be readily attained using Shor's algorithm, Pollard's Rho algorithm, etc (*Aminudin & Cahyono, 2021*; *De Lima Marquezino, Portugal & Lavor, 2019*).

Cryptography algorithms rely on integer mathematics, in particular, number theory to perform invertible operations such as addition, multiplication, exponentiation, etc. over a finite set of integers. Finite Fields, also known as Galois Fields, are fundamental to any cryptographic understanding. A field can be defined as a set of numbers that we can add, subtract, multiply and divide together and only ever end up with a result that exists in our set of numbers. This is mainly advantageous in cryptography since we can only work with a small number of incredibly huge numbers (*Kohli, 2019*). When cryptography algorithms rely solely on converting raw string data in ASCII format, we are restricted to 256 different characters only. Doing such leaves us with only a handful amount of invertible operations in modulo 256 On the other hand, the Galois Field $GF(2^8)$ offers numerous such operations. In fact, Advanced Encryption Standard (AES) (*Daemen & Rijmen, 2001*) uses the multiplicative inverse in $GF(2^8)$. Using the Galois Field also shines forth the opportunity to use the concepts of irreducible polynomials (*Shoup, 1990*). In AES, addition and subtraction is a simple XOR operation. For multiplication, it uses the product modulo an irreducible polynomial. For example, the integer 283 refers to the irreducible polynomial $f(x) = x^8 + x^4 + x^3 + x + 1$ in $GF(2^8)$ whose coefficients are in $GF(2)$ (*Desoky & Ashikhmin, 2006*).

Threshold cryptography is a form of security lock where private keys are distributed among multiple clients or systems. They are even asked to provide digital signature authentication for verification purposes. Only when these keys are combined, can information be effectively decrypted. In practice, this lock is an electronic cryptosystem that protects confidential information, such as a bank account number or an authorization to transfer money from that account (*Henderson, 2020*). The encryption scheme described in this paper has traits that resemble threshold cryptography. Existing threshold cryptosystem protocols might benefit from the positive aspects of our system, making it a viable alternative contender soon. The suggested approach can also be integrated into intelligent systems that use master–slave communication topologies, such as swarm robots (*Chen & Ng, 2021*).

The technique suggested in this study uses an inventive trapdoor function based on the finite field to handle data encryption in cases with enormous string lengths in a reasonable amount of time, demonstrating that it is extremely light. This is a self-adjusting cryptosystem that optimizes the system parameters on its own, saving the user time and

effort while increasing security. The inherent lightness of this cryptosystem makes it an ideal contender for applications involving IoT devices with limited computational power. Confusion and diffusion, covered in 'Observed Security Features', are two aspects of a safe cipher's functioning in cryptography. Due to the system's demonstration of confusion and diffusion properties, it could potentially be used in scenarios such as encrypting bank transaction details, where a high degree of variance in the ciphertext is desirable upon altering few characters in the plaintext. Furthermore, data from our benchmarks in section 'Experimental Analysis' shows promising results when tested on large chunks of data proving that given sufficient computing power, this system could potentially be used for confidential military applications or as a layer of security for the compilation of large datasets in Big Data analytics.

The remainder of this paper is organized as follows. "Literature Review" gives a brief description of numerous sectors where the proposed system can be introduced. "Trapdoor Function" section explains in brief, the working of a traditional trapdoor function from a mathematical perspective. 'Prime number theorem', 'Galois Field in Cryptography', 'Generating Upper Bound for q' and 'Fermat's Factorization' describes the required preliminaries for a better understanding of the algorithm that follows in 'Proposed Algorithm'. Next section talks about two essential properties of the operation of a secure cipher, before moving onto 'Experimental Analysis'. Next, a few ways is covered in which an adversary might try to break into systems running this cryptosystem. 'From an interceptor's perspective' shows that it would be near impossible for them to achieve their goal. 'Remarks on edge cases' addresses an edge case of the system that revolves around the inbuilt block size optimization function. The paper concludes by briefly summarizing the study's overall accomplishments and providing important insights into future research directions.

## LITERATURE REVIEW

Recently in the field of Internet of Things (IoT), research has been conducted on flexible privacy-preserving data publishing schemes in the sector of smart agriculture. Their study shows that over the years protection and privacy concerns for smart agriculture have grown in importance. In these IoT-enabled systems, the internet is used for communicating with participants. Since the cloud is often untrustworthy, higher privacy standards are needed (*Song et al., 2020*).

Security and privacy at the physical layer have become a serious challenge in recent years for numerous communication technologies. IoT networks are typically comprised of a network of interconnected sensors and information relaying units that communicate in real-time with one another. Individual nodes typically have specialized sensor units for detecting specific environmental attributes and have fewer computing resources available. For example, in a house, various technologies such as facial recognition, video monitoring, smart lighting, and so on will all function in tandem. Security and privacy are key impediments to the realistic deployment of smart home technologies (*Shen et al., 2018*).

The majority of the network's elements use sensitive user data and seamlessly exchange information with one another in real-time. To keep intruders out of such a network,

a dependable and stable solution based on edge computing is preferred. Another real-world application of secure edge computing lies in the domain of smart grids. Smart grids are recognized as the next-generation intelligent network that maximizes energy efficiency (*Wang et al., 2020*). Smart grid solutions help to monitor, measure and control power flow in real-time that can contribute to the identification of losses, and thereby appropriate technical and managerial actions can be taken to prevent the same. Smart grids generally rely on data recorded by energy meters from different houses. Since electricity usage data can be classified as confidential user metrics, there is a need for implementing a layer of security before transmitting this data to other parties for further analysis. Encryption of this data at the smart energy meter stage itself can be beneficial. However, this requires the development of a lightweight encryption protocol that can be easily integrated with microprocessors with minimal compute power.

The amount of data provided by users during numerous online activities has increased dramatically over the last decade. Celestine Iwendi et al. performed research that used a model-based data analysis technique for handling applications with Big Data Streaming to glean useful information from this massive amount of data. The method suggested in their research has been tested to add value to large text data processing (*Iwendi et al., 2019*). Our proposed schematic leverages an ingenious trapdoor function based on the finite field to handle data encryption in scenarios involving large string lengths within feasible runtime, proving it to be considerably lightweight.

## TRAPDOOR FUNCTION

The essence of any cryptosystem relies on some special mathematical trapdoor function that makes it practically impossible for an unwelcome interceptor to gain access to secretive information. Simultaneously, these functions also ensure that the authorized parties (who know the secret key) can continue sharing data among themselves.

A trapdoor function is a mathematical transformation that is easy to compute in one direction, but extremely difficult (practically impossible) to compute in the opposite direction in feasible runtime unless some special information is known (private key). Analogously, this can be thought of like the lock and key in modern cryptography where until and unless someone has access to the exact key, they can't open the lock. In mathematical terms, if $f$ is a trapdoor function, then $y = f(x)$ easy to calculate but $x = f^{-1}(y)$ is tremendously hard to compute without some special knowledge $k$ (called key). In case $k$ is known, it becomes easy to compute the inverse $x = f^{-1}(x, k)$.

The components of the proposed system in this paper that act as the trapdoor function is the modulo operation on a Galois field (*Benvenuto, 2012*).

## PRIME NUMBER THEOREM

Positive integers that are divisible by 1 and itself, are known as prime numbers. The sequence begins like the following...

$$2, 3, 5, 7, 11, 13, 17, 19, 23, 29, 31, 37, \ldots$$

**Table 1  Prime density and approximation to logarithmic integral.**

| Search Size $x$ | # of Primes | Density (%) | li($x$) | $\frac{li(x)-\pi(x)}{\pi(x)} \times 100$ |
|---|---|---|---|---|
| 10 | 4 | 40 | 6.16 | 54.14 |
| $10^2$ | 25 | 25 | 30.13 | 20.50 |
| $10^3$ | 168 | 16.8 | 177.61 | 5.72 |
| $10^4$ | 1229 | 12.3 | 1246.14 | 1.39 |
| $10^5$ | 9592 | 9.6 | 9629.81 | 0.39 |
| $10^6$ | 78498 | 7.8 | 78625 | 0.17 |
| $10^7$ | 664579 | 6.6 | 664918 | 0.05 |
| $10^8$ | 5761455 | 5.8 | $5.76 \times 10^6$ | 0.01 |

and has held untold fascination for mathematicians, both professionals and amateurs alike. A result that gives an idea about an asymptotic distribution of primes is known as the prime number theorem (*Goldstein, 1973*).

$\pi(x)$ is the prime-counting function that gives the number of primes less than or equal to $x$, for any real number $x$. This can be written as

$$\pi(x) = \sum_{p \leq x} 1 \tag{1}$$

It is seen *via* graphing, that $\frac{x}{\ln x}$ is a good approximation to $\pi(x)$, in the sense that the limit of the quotient of the two functions $\pi(x)$ and $\frac{x}{\ln x}$ as $x$ increases without bound is 1.

$$\lim_{x \to \infty} \pi(x) \frac{\ln x}{x} = 1 \tag{2}$$

This result can be rewritten in asymptotic notation as

$$\pi(x) \sim \frac{x}{\ln x} \tag{3}$$

The logarithmic integral provides a good estimate to the prime density function.

$$\frac{\pi(x)}{x} \sim li(x) = \int_2^x \frac{1}{\ln t} \, dt \tag{4}$$

To get an idea of the distribution of primes, it is important to count the number of primes in a given range and find the percentage of primes. Consider an infinitely tall tree, losing its leaves. The leaves represent prime numbers. Most leaves are found near the root, and the number of leaves reduces as we walk away from the center. But no matter how far we are from the center, we always find more leaves. These leaves are unpredictably scattered in an infinite area surrounding the tree. This is the situation with the distribution of primes as resembled by Table 1. Figure 1 shows the prime density and logarithmic integral on the left, and the asymptotic nature of the prime counting function on the right.

## GALOIS FIELD IN CRYPTOGRAPHY

Galois Field, named after Evariste Galois, also known as finite field, refers to a field in which there exist finitely many elements. A computer only understands the binary data

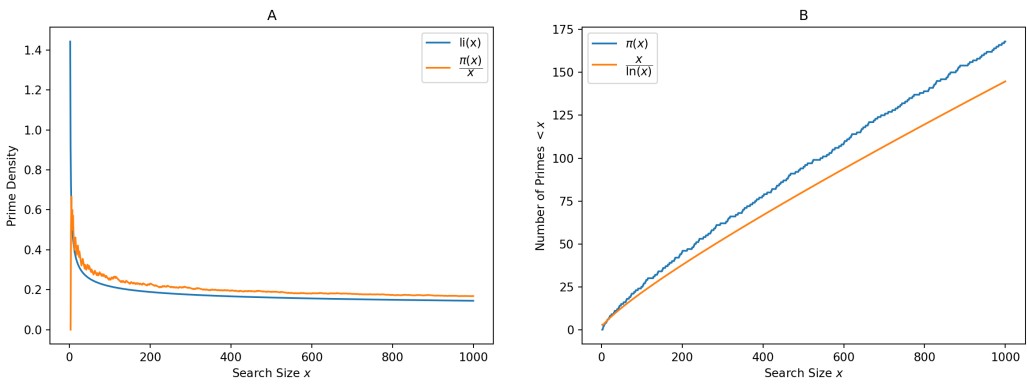

**Figure 1** **Demonstration of the Prime Number Theorem.** (A) Shows Prime density and logarithmic integral while (B) resembles the asymptotic form of prime counting function.

format, which consists of a combination of 0's and 1's. If we consider $GF(2)$, which is simply the Galois Field of order 2, this representation becomes possible which enables us to apply mathematical operations for functional data scrambling. The elements of Galois Field $GF(p^n)$ is defined as

$$
\begin{aligned}
GF(p^n) = & (0, 1, 2, \ldots,) \cup \\
& (p, p+1, p+2, \ldots, p+p-1) \cup \\
& (p^2, p^2+1, \ldots, p^2+p-1) \cup \cdots \cup \\
& (p^{n-1}, p^{n-1}+1, p^{n-1}+2, \ldots, p^{n-1}+p-1)
\end{aligned}
$$

where $p \in \mathbb{P}$ and $n \in \mathbb{Z}^+$. The order of the field is given by $p^n$ while $p$ is called the characteristic of the field. On the other hand, $GF$, as one may have guessed it, stands for Galois Field. Also note that the degree of polynomial of each element is at most $n-1$ (*Benvenuto, 2012*).

## Properties of finite field

For arbitrary elements $a, b, c$ and binary operations $(+, \cdot)$ in a finite field $\mathbb{F}$, the following properties hold (*Stallings, 2006*; *Kohli, 2019*).

1. *Closure:* For any two elements $a, b$, $a+b \in \mathbb{F}$, $a \cdot b \in \mathbb{F}$
2. *Associativity:* $(a+b)+c = a+(b+c), (a \cdot b) \cdot c = a \cdot (b \cdot c)$
3. *Commutativity:* $a+b = b+a, a \cdot b = b \cdot a$
4. *Identity:* There exists a 0 such that for any element $a$ in the field $a+0 = 0+a = a$ known as *additive identity*. There exists a 1 such that for any element $a$ in the field $a \cdot 1 = 1 \cdot a = a$ *multiplicative identity*.
5. Every arbitrary element $a$ has an *additive inverse* $a^{-1}$ such that $a+a^{-1} = a^{-1}+a = 0$ and a *multiplicative inverse* $a^{-1}$ such that $a \cdot a^{-1} = a^{-1} \cdot a = 1$

## Finite field operations

Let $f(p)$ and $g(p)$ two polynomials in the Galois field $GF(p^n)$ with the respective coefficients $A = a_0, a_1, \ldots, a_n$ and $B = b_0, b_1, \ldots, b_n$ for $f, g$. Then the following operations are valid

1. *Addition and Subtraction*

   $$c_k \equiv a_k \pm b_k \bmod p \tag{5}$$

2. *Multiplication and Multiplicative Inverse* For an irreducible polynomial $m(p)$ with a degree of at least $n$, we have the following

   $$h(p) \equiv f(p) \cdot g(p) \bmod m(p) \tag{6}$$

   and polynomials $x(p), y(p)$ are called multiplicative inverses of each other iff

   $$x(p)y(p) \equiv 1 \bmod m(p) \tag{7}$$

## Applications in cryptography

Cryptography is the most prominent and extensively used application of Galois Field. There are many different representations of data. One such representation is a vector in a finite field. Once the data is in this desired format, finite field arithmetic easily facilitates calculations during encryption and decryption (*Benvenuto, 2012*). In the 1970's, IBM developed Data Encryption Standard (DES) (*Tuchman, 1997*). However, a humble 56-bit key usage never posed a serious challenge to a supercomputer, which was able to break the key in less than 24 hours. Thus the need for a refined algorithm to replace the existing DES arise. Rijndael, a much more sophisticated algorithm devised by Vincent Rijmen and John Daemon in 2001, has been known as the Advanced Encryption Standard (AES) ever since. An issue regarding this breakthrough was published by Federal Information Processing Standards Publications (FIPS) on November 26, 2001 (*Dworkin et al., 2001*).

## GENERATING UPPER BOUND FOR Q

The proposed algorithm, requires generating a list of primes $p_s \in \mathbb{P}$ whose size equals the block size $s$. It also requires a prime $q \in \mathbb{P}$ which is greater than all $p_s$'s. The algorithm involves generating inverses of all $p_s$'s within a Galois field $GF(q^m)$ where $m$ is an arbitrarily chosen positive integer.

Generation of the shuffled list of primes requires us to know the prime $q$. On the other hand, determining the value of $q$ requires us to know the largest prime present in the shuffled list. This poses a paradoxical problem.

To get around this paradox, we consider the following.

- The number of primes to be stored in the shuffled list $p_s$ is equivalent to the block size $s$.
- The value $q$ must be chosen such that it is prime and larger than the maximum prime present in $p_s$.
- Prime Number Theorem is used to find is used to find the number below which $s$ primes are available. Let this required number be denoted by $x$.

$$s = \frac{x}{\ln x}$$

$$s \ln x = x$$

$$s \ln x = e^{\ln x}$$

$$\ln x e^{-\ln x} = \frac{1}{s}$$

$$-\ln x e^{-\ln x} = -\frac{1}{s}$$

$$-\ln x = W_n\left(-\frac{1}{s}\right)$$

$$x = e^{-W_n\left(-\frac{1}{s}\right)}$$

where $W_n$ is the Lambert W Function also known as the product log function. This means that it is possible to generate the upper limit for generating a list of primes for an arbitrary block size. Since the number of primes is positive, $x$ must be positive (*Weisstein, 2002*; *Corless et al., 1996*). This means that it is possible to have a Lambert W Function in the branch of $n = 0$ or $n = -1$ since $e^y \geq 0 \forall y \in \mathbb{R}$. If $x = a + ib$ that is $x \in \mathbb{C}$, we consider $\lfloor \Re(x) \rfloor$ to generate the upper bound.

- The prime larger than this $x$ is set to be $q$.

## FERMAT'S FACTORIZATION

Fermat's factorization method, named after Pierre de Fermat, is based on the representation of an odd integer as the difference of two squares (*De Fermat, 1891*). For a given number $N$, the objective is to find $a, b$ such that

$$N = a^2 - b^2 \tag{8}$$

To start, the square root of $N$ is taken, and the nearest integer $a$ is squared and subtracted. If the resulting number is a square then, $a, b$ has been found. If it is not the case, then $a$ is increased by 1 and the process is repeated. This is what is used to generate an algorithm for the block size depending on the size of the plaintext.

## OPTIMAL CHOICE FOR BLOCK SIZE

The following is an algorithm that automatically chooses an optimal value for the block size so as to ensure minimum time of execution for the algorithm.

1. Get the length $N$ of plaintext. Take square root and consider the ceiling of the resulting real number, i.e., $\left\lfloor \sqrt{N} \right\rfloor$

2. If $\left(\left\lfloor \sqrt{N} \right\rfloor\right)^2 \leq N$ and $N \equiv 1 \bmod 2$, then optimal block size is $\left\lfloor \sqrt{N} \right\rfloor$

## PROPOSED ALGORITHM

The proposed algorithm for this cryptosystem involves numerous sections, making it a robust and impenetrable layer of security. Note that this algorithm autonomously sets the critical private key parameters of the system to their optimal values based on the user's secret message. Figure 2 illustrates a flow chart for the proposed schematic.

For IoT-enabled networking devices, an additional layer of intrusion detection protocol can be appended to the proposed scheme to enhance the existing security. The Internet of

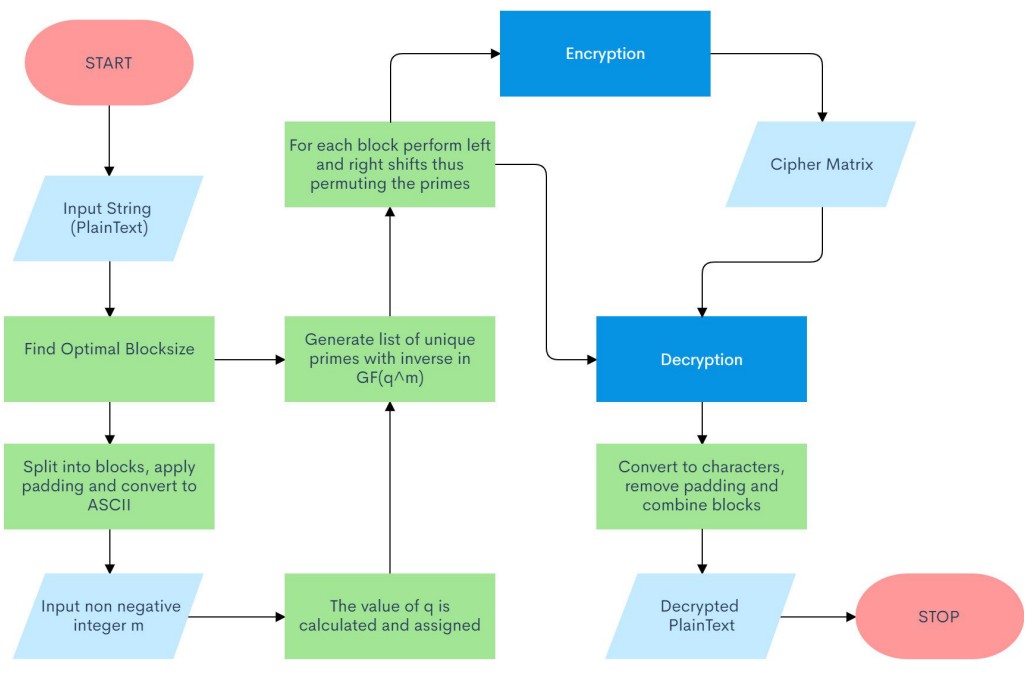

**Figure 2  Algorithm Flow Diagram.**

Medical Things (IoMT) is a subset of IoT in which medical equipments connect to share sensitive data. In such scenarios, machine learning methods are commonly employed in Intrusion Detection Systems (IDS) for dynamically identifying and categorizing threats at the network and host levels (*Swarna Priya et al., 2020*).

## Plaintext pre-processing

1. An input message is provided and split into characters.
2. The optimum block size is evaluated as discussed in section 'Optimal choice for block size'.
3. The plaintext is split into blocks and padding is applied to maintain consistency of the modified plaintext.
4. Each character in each block is converted to its designated ASCII equivalent.

## Key generation

1. User inputs a non negative integer $m$, which is used to determine the Galois Field $GF(q^m)$.
2. The value of $q$ is obtained *via* the calculation described in section 'Generating upper bound for q'.
3. A function sets an optimal block size depending on the length of plaintext *via* the algorithm described in section 'Optimal choice for block size'.
4. A list of unique primes is randomly generated between $[p_1, p_{\text{blocksize}}]$ and $p_k^{-1}$ mod $q^m$ exists, such that each block has the same primes. The size of this list is equal to the block size.

### Key permutation algorithm

1. Since each block is assigned the same primes, entropy is introduced into the system by rearranging the order of primes for each block in the list of primes.
2. A central element is kept fixed, the primes on the left side are left-shifted and the ones on the right are right-shifted a certain number of times. The shift factor consequently increases linearly as the block index increases. It is crucial that the block size must be an odd number for such a rearrangement procedure to take place.

### Encryption

For each block, take the product of the corresponding ASCII values $a_k$ and the prime number from the permuted prime list. Then take this product modulo $q^m$.

$$a_k p_k \equiv c_k \ \text{mod} \ q^m \tag{9}$$

These lists of $c_k$ are arranged in a matrix of order $\frac{n}{s} \times s$, where $n =$ length of padded plaintext and $s =$ block size.

### Decryption

For each encrypted value $c_k$ in the block, multiply with the inverse of corresponding prime in the permuted list in the field $q^m$.

$$c_k p_k^{-1} \equiv a_k \ \text{mod} \ q^m \tag{10}$$

Padding is removed, and the remaining characters are joined to return the original message.

## OBSERVED SECURITY FEATURES

In cryptography, confusion and diffusion are two properties of the operation of a secure cipher which were identified by Claude Shannon in his paper Communication Theory of Secrecy Systems, published in 1949 (*Shannon, 1949*).

*Confusion* is a technique that ensures confidentiality, that is, a ciphertext gives no clue about the plaintext. This is commonly used in the block and stream cipher method. This can be achieved by the substitution method.

An extensive analysis was performed to study the behavior of the encryption scheme when the same plaintext was encrypted twice with only 1 single character changed. Two simple messages ('abcdefghijklmnopqrstuvwxyz') and ('abcdefghijklmnopqrstuvwxyp') were considered. Note that both messages are identical except for one single character at the end ('z' and 'p'). An identical set of encryption parameters were set up for this analysis and the generated ciphertext in both cases were noted down:

$$\text{abcdefghijklmnopqrstuvwxyz} \xrightarrow[m=7]{\text{Encryption}} \begin{bmatrix} 679 & 1078 & 198 & 500 & 303 \\ 1122 & 721 & 208 & 315 & 530 \\ 749 & 1188 & 218 & 550 & 333 \\ 1232 & 791 & 228 & 345 & 580 \\ 819 & 1298 & 238 & 600 & 363 \\ 854 & 528 & 96 & 240 & 144 \end{bmatrix}$$

$$\text{abcdefghijklmnopqrstuvwxyp} \xrightarrow[m=7]{\text{Encryption}} \begin{bmatrix} 194 & 686 & 1089 & 300 & 505 \\ 714 & 206 & 1144 & 525 & 318 \\ 214 & 756 & 1199 & 330 & 555 \\ 784 & 226 & 1254 & 575 & 348 \\ 234 & 826 & 1309 & 360 & 605 \\ 224 & 336 & 528 & 144 & 240 \end{bmatrix}$$

If each element of both the ciphertext matrices are compared element-wise (order matters), one can easily notice that there is 0% similarity. This implies that cryptanalysis techniques that rely on the similarity of elements in ciphertexts will fail to crack this cryptosystem.

In *diffusion*, the statistical structure of the plaintext is dissipated into long-range statistics of the ciphertext (*Stallings, 2006*). This increases the redundancy of the plaintext by spreading it across rows and columns. It is only used in block cipher protocols. This phenomenon can be achieved by a permutation technique known as Transposition. A perfect example of diffusion and confusion is the AES cryptosystem.

Additionally, the encryption scheme was also tested with strings which have the same character repeating multiple time. For instance, the plaintext ('she sells sea shells on the sea shore') was encrypted using regular encryption parameters. The following ciphertext matrix was returned by the algorithm:

$$\text{she sells sea shells on the sea shore} \xrightarrow[m=3]{\text{Encryption}} \begin{bmatrix} 345 & 208 & 505 & 352 & 805 \\ 202 & 324 & 540 & 805 & 352 \\ 345 & 202 & 485 & 352 & 805 \\ 208 & 303 & 540 & 756 & 1265 \\ 96 & 222 & 550 & 352 & 812 \\ 312 & 202 & 160 & 1265 & 707 \\ 194 & 96 & 575 & 728 & 1221 \\ 342 & 202 & 240 & 528 & 336 \end{bmatrix}$$

From an unauthorized eavesdropper's perspective, the ciphertext matrix will give the impression of being just a random sequence of numbers which makes it all the more difficult to come up with a logical approach to retrieve the secret message without any knowledge of the private key.

## EXPERIMENTAL ANALYSIS

### Stringlength *vs* time for encrypt-decrypt cycle

For this benchmark test, the value of $m$ was fixed at 7 different values $m = 2^n$ where $n \in [0, 6], n \in \mathbb{Z}$ while the length of plaintext was successively increased in powers of 10 starting from 100 till 1,000,000 while noting down the time it takes for successful encrypt-decrypt cycles. The tabular data shown in Tables 2 and 3 shows the variance of runtime upon altering string lengths. Figure 3 shows the implementation of the program on a Intel® Core™ i7-10750H CPU®2.60 GHz and a Raspberry Pi 4 Model B (Quad core Cortex-A72 (ARM v8) 64-bit SoC®1.5 GHz) respectively.

**Table 2** Effect of length of plaintext on runtime (seconds) for different values of *m* for Intel® Core™ i7-10750H.

| | Values for *m* | | | | | | |
|---|---|---|---|---|---|---|---|
| String Length | 1 | 2 | 4 | 8 | 16 | 32 | 64 |
| $10^2$ | 0.0005 | 0.0002 | 0.0002 | 0.0003 | 0.0004 | 0.0007 | 0.0004 |
| $10^3$ | 0.0024 | 0.0024 | 0.0024 | 0.0027 | 0.0030 | 0.0037 | 0.0038 |
| $10^4$ | 0.0311 | 0.0393 | 0.0321 | 0.0342 | 0.0358 | 0.0418 | 0.0470 |
| $10^5$ | 0.3143 | 0.3201 | 0.3596 | 0.3580 | 0.3975 | 0.4288 | 0.5501 |
| $10^6$ | 3.2605 | 3.3822 | 3.9073 | 4.0346 | 4.4580 | 5.0636 | 6.0518 |

**Table 3** Effect of length of plaintext on runtime (seconds) for different values of *m* for Raspberry pi.

| | Values for *m* | | | | | | |
|---|---|---|---|---|---|---|---|
| String length | 1 | 2 | 4 | 8 | 16 | 32 | 64 |
| $10^2$ | 0.0010 | 0.0011 | 0.0012 | 0.0012 | 0.0013 | 0.0015 | 0.0018 |
| $10^3$ | 0.0101 | 0.0102 | 0.0116 | 0.0122 | 0.0138 | 0.0160 | 0.0205 |
| $10^4$ | 0.1176 | 0.1248 | 0.1379 | 0.1399 | 0.1587 | 0.1911 | 0.2529 |
| $10^5$ | 1.3059 | 1.4381 | 1.4981 | 1.6747 | 1.8104 | 2.1556 | 2.9447 |
| $10^6$ | 14.3205 | 16.3886 | 16.7864 | 18.4380 | 20.6584 | 24.5239 | 34.1969 |

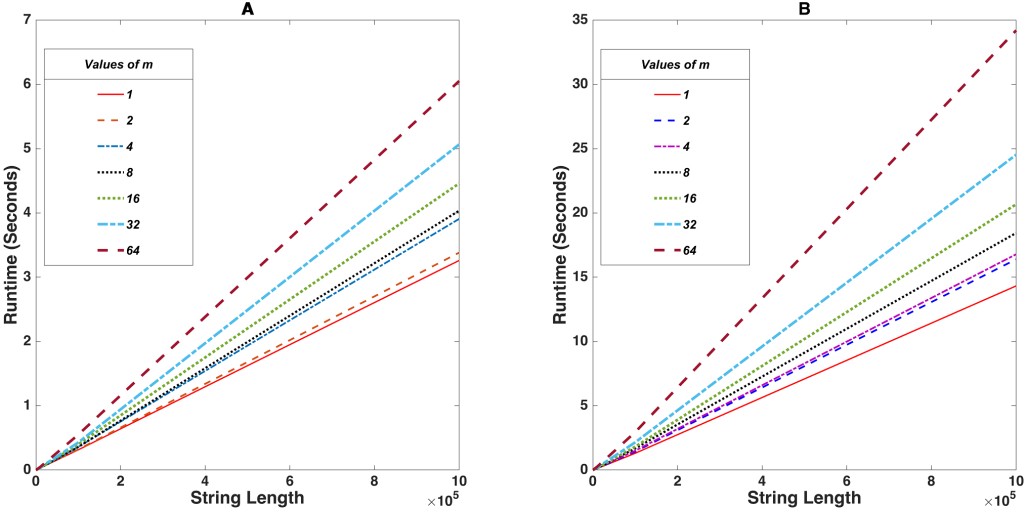

**Figure 3** **Stringlength *vs* Runtime (seconds) for fixed values of *m*.** (A) and (B) shows the demonstration on an Intel® Core™ i7-10750H and a Raspberry Pi 4 Model B respectively.

Here *m* is the exponent of a prime finite field. It means that any arbitrary value *a* mod *q* will generate non-negative integers within $[0, q-1]$. But for any $m > 0, m \in \mathbb{Z}$, we have $q^m > q$ which consequently means *a* mod $q^m$ generates non negative integers within $[0, q^m - 1]$ which gives larger values. This takes a bit of time to process. Hence smaller values of *m* will be less impactful on the time constraint.

**Table 4  Effect of exponent of prime finite field _m_ on runtime for fixed string (size = 1000).**

| Values of _m_ | Runtime (seconds) | |
| --- | --- | --- |
| | Intel® Core™ i7-10750H | Raspberry Pi 4 Model B |
| 1 | 0.0025 | 0.0099 |
| 2 | 0.0025 | 0.0010 |
| 4 | 0.0024 | 0.0111 |
| 8 | 0.0025 | 0.0120 |
| 16 | 0.0028 | 0.0134 |
| 32 | 0.0035 | 0.0160 |
| 64 | 0.0036 | 0.0204 |
| 128 | 0.0054 | 0.0301 |
| 256 | 0.0076 | 0.0556 |
| 512 | 0.0153 | 0.1294 |
| 1024 | 0.0416 | 0.3402 |

To study the behavior of this algorithm on devices with low compute power, it was benchmarked on a Raspberry Pi 4 Model B (8 GB RAM variant). The Raspberry Pi is a low-cost, credit-card-sized device that connects to a computer monitor or TV and operates with a regular keyboard and mouse. It is sometimes referred to as a Single Board Computer (SBC) because it runs a complete operating system and has enough peripherals (memory, Processor, power regulation) to begin execution without the inclusion of hardware. The Raspberry Pi can run various operating systems and needs only power to boot. Some Raspberry Pi models can boot directly from the network, but in general, file-system storage, such as a micro SD card, is necessary (_Johnston & Cox, 2017_). The Raspberry Pi features GPIO (general purpose input/output) pins that allow one to manipulate electronic components and low-powered sensors for physical computing and explore the Internet of Things.

In this case, it was observed that if we limit the string length to 20,000 characters, the encryption-decryption cycle completes within a mere 5 seconds. Altering the m-values and repeating the benchmark made a negligible difference, as seen on the graph. It should be noted that executing this benchmark on the raspberry pi for 100,000 characters takes up to 35 seconds or more. However, most IoT applications involve the collection of data from various sensors and transmitting them in discrete chunks to servers across multiple timesteps for further processing. In such scenarios dealing with limited batches of data, the proposed cryptosystem can achieve feasible encryption in real-time.

## Variation of runtime with value of m

This section analyzes how changing the exponent of the prime finite field _m_ has an influence on the operational runtime of the encrypt and decrypt functions when a fixed string of random alphanumeric values having a size of 1000 characters is fed into the proposed algorithm. The algorithm autonomously sets the block size to 31. This is clearly visible from the data in Table 4.

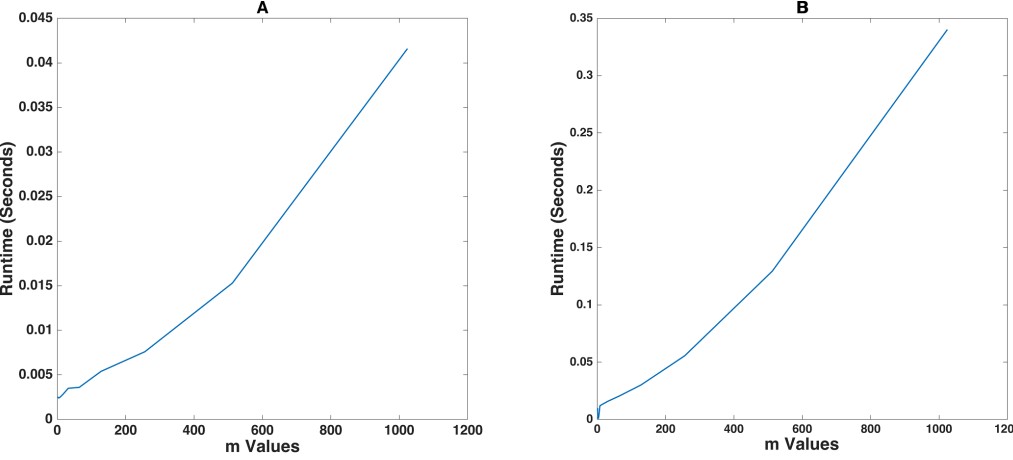

**Figure 4** **Prime finite field exponent *m* vs Runtime for plaintext of 1000 characters.** (A) and (B) shows the demonstration on an Intel® Core™ i7-10750H and a Raspberry Pi 4 Model B respectively.

Figure 4 demonstrates that *m* vs runtime follows a fairly linear trend. There is a slight imperfection in the trend since at this magnified time scale, changes in the memory usage patterns of the system can lead to noticeable changes in the efficiency of the algorithm. Note that these are of the order of $10^{-3}$ seconds when executed on a workstation. Testing this on the Raspberry Pi, however, took 10 times more time than that on the computer.

# FROM AN INTERCEPTOR'S PERSPECTIVE

## Longer messages, greater security

The proposed system has a built-in recommendation system due to its design of the algorithm. It automatically sets the block size depending on the length of the input message. It has been noted that the block size increases as the length of the input string increases. A larger block size means each block has more primes.

If block size $= n$, then there are $n!$ arrangements possible. Only one of them is the correct combination for each block. So an interceptor has a $\frac{1}{n!}$ chance to get the combination right, for successful decoding. Since

$$\lim_{n \to \infty} \frac{1}{n!} = 0 \tag{11}$$

it consequently means that a larger block size makes it nearly impossible for an attacker to correctly guess all the permuted blocks.

## Hensel's Lifting Lemma

**Lemma 1 (Hensel's Lemma):** *Given prime $p, e \geq 2$, and $f(x) \in \mathbb{Z}[x]$, if $a$ is a solution to $f(x) \equiv 0 \pmod{p^{e-1}}$.*

*Then if $\gcd(p, f'(a)) = 1$, there exists a solution to $f(x) \equiv 0 \pmod{p^e}$ of the form $b = a + kp^{e-1}$ where $k$ satisfies*

$$\frac{f(a)}{p^{e-1}} + kf'(a) \equiv 0 \pmod{p} .$$

This cryptosystem requires the user to pick an exponent $m$ for an automatically generated $q$ value, allowing the use of the field of order $q^m$. When it comes to an interceptor, they only have access to a number $b$ where $b = q^m$, and have no idea of $q$ and $m$ separately. Hence, they would have to apply heuristic, or brute force approach to solve for $q$ and $m$ given the value of $b$. This is because there are no known, deterministic methods to solve an equation with two unknowns.

## REMARKS ON EDGE CASES

An imperative consequence of the block size optimization function described in section 'Optimal choice for block size' is that

No. of Blocks $\geq$ Block size

The key generation paradigm demands that the size of the final list of unique primes should be the same as the block size and all elements of this list should be smaller than the value of $q$ that is evaluated in the background using the prime number theorem as discussed in section 'Generating upper bound for q'. This implies that if a plaintext with very few characters is chosen such that Block size $\leq 3$, the shuffled list of unique random primes can hold only 2 possible elements *i.e.,* $2, 3$ which are both lesser than the calculated value of $q = 5$. In this scenario, since 3 primes are not available, the system gets hung up in an infinite loop and fails to encrypt the message. For instance, when the message "hello" was passed, it resulted in the optimal block size being set to 3 and $q = 5$. Irrespective of the value of the prime finite field exponent $m$ chosen by the user, it was observed that the key generation algorithm breaks down.

One simple way to solve this issue would be to add a different special padding scheme in case the message entered by the user is too small to ensure that the optimal block size evaluates to a number greater than or equal to 5. This way, the key generation algorithm has enough prime numbers available to work with.

## CONCLUSION

In this paper, a new block cipher encryption scheme was discussed in detail. It was observed that longer messages provide better security whereas shorter messages provide faster execution assuming sufficient padding. This system can come in handy, especially in social media sites where the short messaging system (SMS) is common. For example, Twitter, which has a current maximum string length of 280 characters (*Twitter, 2021*). The time of execution was benchmarked on a modern-day computer CPU (Intel® Core™ i7-10750H processor) as well as on a Raspberry Pi 4 Model B. It was found that the proposed schematic can easily be integrated into IoT networks involving low compute microprocessors to provide a layer of security. Other applications of this system could be in encrypting confidential military files that are large. As a threshold cryptosystem candidate, this system can find multiple applications in swarm robotics in cases where slaves communicate with a master robot over an insecure network. The list of permuted primes that constitutes the private key of this system could be scattered across multiple

slaves and could be used collectively to ensure that none of the nodes in the system gets attacked by an unauthorized party and/or fails at any given time. Whether or not this could be used as an industry standard is beyond the scope of this paper. The progress so far has been compiled into a GitHub repository (*Bhowmik & Menon, 2020*).

## ACKNOWLEDGEMENTS

We are extremely thankful to Debwashis Borman for his assistance with plotting the obtained results in MATLAB.

### Funding

The authors received no funding for this work.

### Competing Interests

The authors declare there are no competing interests.

### Author Contributions

- Awnon Bhowmik and Unnikrishnan Menon conceived and designed the experiments, performed the experiments, analyzed the data, performed the computation work, prepared figures and/or tables, authored or reviewed drafts of the paper, and approved the final draft.

### Data Availability

The code and raw results are available at GitHub: https://github.com/awnonbhowmik/Adaptive-Cryptosystem-Finite-Field.

### Supplemental Information

Supplemental information for this article can be found online at http://dx.doi.org/10.7717/peerj-cs.637#supplemental-information.

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
