# Peer review of "An adaptive cryptosystem on a Finite Field"

_PeerJ Computer Science, doi:10.7717/peerj-cs.637_

## Round 0.1 · original submission · Major Revisions

Dear Authors,

Based on the recommendation from the reviewers, I recommend a major revision for the article. All the comments from the reviewers should be carefully addressed before submission.

Reviewer 1 ·

Basic reporting

The paper introduces an adaptive block encryption scheme. This system is based on product, exponent and modulo operation on a finite field. At the heart of this algorithm lies an innovative and robust trapdoor function that operates in the Galois Field and is responsible for the superior speed and security offered by it. Prime number theorem plays a fundamental role in this system, to keep unwelcome adversaries at bay. This is a self-adjusting cryptosystem that autonomously optimizes the system parameters thereby reducing effort on the user's side while enhancing the level of security. This paper provides an extensive analysis on a few notable attributes of this cryptosystem such as it's exponential rise in security with an increase in the length of plaintext while simultaneously ensuring that the operations are carried out in feasible runtime. Additionally, an experimental analysis is also performed to to study the trends and relations between the different parameters of the cryptosystem including a few edge cases.

Experimental design

no comment

Validity of the findings

no comment

Additional comments

1. Please improve overall readability of the paper.
2. The objectives of this paper need to be polished.
3. Introduction is poorly written.
4. Relevant literature review of latest similar research studies on the topic at hand must be
discussed.
5. Result section need to be polished.
6. There are some grammar and typo errors.
7. Improve the quality of figures
8. Define all the variables before using
The authors can cite the following
1.A Novel PCA-Firefly based XGBoost classification model for Intrusion Detection in Networks using GPU
2. Fake Review Classification Using Supervised Machine Learning

Reviewer 2 ·

Basic reporting

no comment

Experimental design

no comment

Validity of the findings

no comment

Additional comments

In this paper, the authors designed a novel symmetric encryption method based on finite field. Although the proposed protocol has no obvious mistakes and question, some revisions are still needed for this paper. The authors may refer to the following comments for revision.

1. Related work part cannot be found in this paper.

2. No comparative experiments conducted, the authors should compare their proposed protocol with other similiar encryption protocol in terms of computation cost and other metrics.

3. Some typos and grammar errors exist in this paper, the authors should double check the paper.

4. More supplementary references should be used in the manuscript. For authors' convenience, we list the related references as follows.
1. Song J, Zhong Q, Wang W, et al. FPDP: Flexible privacy-preserving data publishing scheme for smart agriculture. IEEE Sensors Journal, 2020, doi: 10.1109/JSEN.2020.3017695.
2. Wang W, Su C. Ccbrsn: a system with high embedding capacity for covert communication in bitcoin//IFIP International Conference on ICT Systems Security and Privacy Protection. Springer, Cham, 2020: 324-337.
3. Wang W, Huang H, Zhang L, et al. Secure and efficient mutual authentication protocol for smart grid under blockchain. Peer-to-Peer Networking and Applications, 2020: 1-13.
4. Zhang L, Zou Y, Wang W, et al. Resource Allocation and Trust Computing for Blockchain-Enabled Edge Computing System. Computers \& Security, 2021: 102249.
5. Zhang L, Zhang Z, Wang W, et al. Research on a Covert Communication Model Realized by Using Smart Contracts in Blockchain Environment. IEEE Systems Journal, doi: 10.1109/JSYST.2021.3057333.

Reviewer 3 ·

Basic reporting

Professional English is required to improve the paper before publication.

Literature review is not sufficient. Latest and related paper like enhancement of the Cryptosystem should be compared with https://ieeexplore.ieee.org/abstract/document/6294370
How Big data and cryptography should be analysed with https://www.mdpi.com/2079-9292/8/11/1331
You can apply optimization approach
https://onlinelibrary.wiley.com/doi/abs/10.1002/spe.2797

Figures need improvement
Equations are not numbered
More clarity in results needed

Experimental design

Research questions need to be refined
Methods okay in mathematical analysis

Validity of the findings

Findings are okay
Conclusion can be improved

Additional comments

Professional English is required to improve the paper before publication.

Literature review is not sufficient. Latest and related paper like enhancement of the Cryptosystem should be compared with https://ieeexplore.ieee.org/abstract/document/6294370
How Big data and cryptography should be analysed with https://www.mdpi.com/2079-9292/8/11/1331

Figures need improvement
Equations are not numbered
More clarity in results needed
Research questions need to be refined
Methods okay in mathematical analysis

---

## Round 0.2 · Minor Revisions

Based on the comments from the reviewers and my own observations, I suggest minor revisions for the paper. Please address their comments.

Reviewer 1 ·

Basic reporting

paper introduces an adaptive block encryption scheme. This system is
based on product, exponent, and modulo operation on a finite field. At the heart of this algorithm lies
an innovative and robust trapdoor function that operates in the Galois Field and is responsible for the
superior speed and security offered by it. Prime number theorem plays a fundamental role in this system,
to keep unwelcome adversaries at bay. This is a self-adjusting cryptosystem that autonomously optimizes
the system parameters thereby reducing effort on the user’s side while enhancing the level of security.
This paper provides an extensive analysis of a few notable attributes of this cryptosystem such as its
exponential rise in security with an increase in the length of plaintext while simultaneously ensuring that
the operations are carried out in feasible runtime. Additionally, an experimental analysis is also performed
to study the trends and relations between the cryptosystem parameters, including a few edge cases

Experimental design

.

Validity of the findings

.

Additional comments

• In Introduction section, the drawbacks of each conventional technique should be described clearly.
• Introduction needs to explain the main contributions of the work more clearly.
• The authors should emphasize the difference between other methods to clarify the position of this work further.
• The Wide ranges of applications need to be addressed in Introductions
• The objective of the research should be clearly defined in the last paragraph of the introduction section.
• Add the advantages of the proposed system in one quoted line for justifying the proposed approach in the Introduction section.

In preprocessing the authors can refer the following
A Novel PCA-Firefly based XGBoost classification model for Intrusion Detection in Networks using GPU
An effective feature engineering for DNN using hybrid PCA-GWO for intrusion detection in IoMT architecture

Reviewer 2 ·

Basic reporting

The authors have well addressed my comments, so it is ready for publication now

Experimental design

The authors have well addressed my comments, so it is ready for publication now

Validity of the findings

The authors have well addressed my comments, so it is ready for publication now

Additional comments

The authors have well addressed my comments, so it is ready for publication now

Reviewer 3 ·

Basic reporting

The Paper is improved but lack recent References

Contribution is still lacking at the end of Introduction

Figures still not clear enough. Some further improvement needed

Experimental design

improved

Validity of the findings

Improved

Additional comments

The Paper is improved but lack recent References
Add
https://www.mdpi.com/1424-8220/20/9/2609

Contribution is still lacking at the end of Introduction

Figures still not clear enough. Some further improvement needed

---

## Round 0.3 · accepted · Accept

Dear Dr. Bhowmik,

Thank you for your submission to PeerJ Computer Science.

I am writing to inform you that your manuscript - An adaptive cryptosystem on a Finite Field - has been Accepted for publication. Congratulations!

Reviewer 1 ·

Basic reporting

The authors have addressed all of my comments, paper can be accepted in the current form. Thank you for giving me this opportunity.

Experimental design

Good

Validity of the findings

Good

Additional comments

The authors have addressed all of my comments, paper can be accepted in the current form. Thank you for giving me this opportunity.

Reviewer 2 ·

Basic reporting

The authors have addressed all the comments. The paper can be accepted for publication.

Experimental design

The authors have addressed all the comments. The paper can be accepted for publication.

Validity of the findings

The authors have addressed all the comments. The paper can be accepted for publication.

Additional comments

The authors have addressed all the comments. The paper can be accepted for publication.

Reviewer 3 ·

Basic reporting

The work has been improved, but in view of the Case study penetration testing models evaluation: how many case studies and how many testers are required?

Experimental design

The work has been improved, but in view of the Case study penetration testing models evaluation: how many case studies and how many testers are required?

Validity of the findings

Improved

Additional comments

The work has been improved, but in view of the Case study penetration testing models evaluation: how many case studies and how many testers are required?